# Selective Inhibition of Genomic and Non-Genomic Effects of Thyroid Hormone Regulates Muscle Cell Differentiation and Metabolic Behavior

**DOI:** 10.3390/ijms22137175

**Published:** 2021-07-02

**Authors:** Annarita Nappi, Melania Murolo, Serena Sagliocchi, Caterina Miro, Annunziata Gaetana Cicatiello, Emery Di Cicco, Rossella Di Paola, Maddalena Raia, Lucia D’Esposito, Mariano Stornaiuolo, Monica Dentice

**Affiliations:** 1Department of Clinical Medicine and Surgery, University of Naples Federico II, 80131 Naples, Italy; annarita.nappi@unina.it (A.N.); Mel.murolo@studenti.unina.it (M.M.); serena.sagliocchi@unina.it (S.S.); caterina.miro@unina.it (C.M.); Annunziatagaetana.cicatiello2@unina.it (A.G.C.); emery.dicicco@unina.it (E.D.C.); ross.dipaola@studenti.unina.it (R.D.P.); 2CEINGE–Biotecnologie Avanzate Scarl, 80131 Naples, Italy; raia@ceinge.unina.it; 3Centro Servizi Veterinari, University of Naples Federico II, 80131 Naples, Italy; lucia.desposito@unina.it; 4Department of Pharmacy, University of Naples Federico II, 80131 Naples, Italy; mariano.stornaiuolo@unina.it

**Keywords:** thyroid hormone, genomic and non-genomic action, deiodinase

## Abstract

Thyroid hormones (THs) are key regulators of different biological processes. Their action involves genomic and non-genomic mechanisms, which together mediate the final effects of TH in target tissues. However, the proportion of the two processes and their contribution to the TH-mediated effects are still poorly understood. Skeletal muscle is a classical target tissue for TH, which regulates muscle strength and contraction, as well as energetic metabolism of myofibers. Here we address the different contribution of genomic and non-genomic action of TH in skeletal muscle cells by specifically silencing the deiodinase *Dio2* or the *β3-Integrin* expression via CRISPR/Cas9 technology. We found that myoblast proliferation is inversely regulated by integrin signal and the D2-dependent TH activation. Similarly, inhibition of the nuclear receptor action reduced myoblast proliferation, confirming that genomic action of TH attenuates proliferative rates. Contrarily, genomic and non-genomic signals promote muscle differentiation and the regulation of the redox state. Taken together, our data reveal that integration of genomic and non-genomic signal pathways finely regulates skeletal muscle physiology. These findings not only contribute to the understanding of the mechanisms involved in TH modulation of muscle physiology but also add insight into the interplay between different mechanisms of action of TH in muscle cells.

## 1. Introduction

Thyroid hormones (THs, T3 and T4) are critical endocrine regulators of tissue development and physiology of almost all the cells and tissues in the adult organs [1]. Their mechanism of action has been generally seen as mediated by genomic regulation of target gene transcription by the mean of nuclear hormone receptors (TRs), ligand-dependent transcription factors and members of the nuclear receptor gene superfamily (the so-called canonic or Type 1 mechanism) [2]. The T3-TR complex recruits regulatory complexes with the final occupancy of specific DNA binding regions (TREs, thyroid response elements) [3]. Since the major product of the thyroid gland is thyroxin, T4, an essential step in the activation of genomic action of TH is the conversion of the so-called pro-hormone T4 into the most active form T3, which has higher affinity for the TRs compared to T4. The activation reaction is a dehalogenation catalyzed by three iodothyronine deiodinases (type 1, 2 and 3 deiodinases, D1, D2 and D3). The removal of one iodine moiety from T4 results in the formation of T3 (catalyzed by D1 and D2) or the rT3 (catalyzed by D1 and D3). The final binding of T3 to nuclear TRs determines the transactivation of many TH target genes [4].

Additionally, extranuclear (non-genomic or Type 2) actions of TH have been widely reported in a variety of cells, which are responsible for rapid effects, mediated by different TH- or TR-partner proteins [2]. In this regard, non-genomic actions have been defined as the mechanisms that do not involve the binding of T3 to TRs and their chromatin occupancy [5]. For instance, it has been demonstrated that T4 (and to a lesser extend T3) can bind a membrane receptor belonging to the Integrin family (αVβ3-Integrin), which in turn results in activation of the MAPK signaling cascade and enhanced cell proliferation [6]. Moreover, the T4-αVβ3-Integrin binding fosters cancer angiogenesis in prostate cancer cells and in tumor endothelial cells [7,8]. Notably, the TH binding to αVβ3-Integrin is specifically inhibited by TETRAC (3, 3, 5, 5′-Tetraiodothyroacetic acid), a deaminated form of T4 [9]. These data show that T4 is far more than a pro-hormone for T3 and that non-genomic TH action must be seen as an alternative way of looking at the overall action in target tissues [10]. Additional non-genomic actions of THs include binding of T3 to the p85 subunit of phosphatidyl-inositol 3-kinase (PI3K), which results in activation of the PIK3/Akt pathway and activation of endothelial nitric oxide synthase (eNOS) [11].

A key function of TH is the ability to act as a powerful regulator of cellular energetic dynamics [12]. This ability includes the enhancement of the basal metabolic rate, the up-regulation of protein turnover and the increase of mitochondria biosynthesis [13]. Also in these cases, TH actions are achieved through a combination of gene transcription activation via T3-TR binding to the chromatin, as well as by intracellular activation of signaling pathways that do not involve classical TR-T3 nuclear action.

Taking into account the multiple roles of TH in muscle cell physiology, our aim was to dissect the specific contributions of the genomic (mediated by the D2-dependent T3 activation) and non-genomic (due to the binding of T4 to the αVβ3-Integrin) actions to the final effects of TH in skeletal muscle.

We evaluated the ability of TH to modulate the proliferation/differentiation balance of muscle cells as well as the regulation of the ROS production in contexts of altered genomic or non-genomic action. Our data demonstrated that muscle cell proliferation is inversely regulated by the D2-dependent T3 activation and the αVβ3-Integrin-dependent T4 action. Conversely, both the Integrin signaling and the deiodination signaling induce muscle cell differentiation. The inhibition of both D2 and αVβ3-Integrin led to increased ROS production, in agreement that TH potentiates the expression of ROS scavengers in skeletal muscle [14].

Considering the plethora and sometimes divergent effects that TH exerts in target cells, understanding the mechanisms by which TH regulates skeletal muscle functions could be crucial to comprehend the complex outcomes induced by TH in the target tissues.

## 2. Results

### 2.1. Inactivation of D2 Expression Enhances Cell Proliferation of Muscle Cells

To assess the contribution of D2-mediated TH nuclear activation on the proliferative ability of skeletal muscle cells, we silenced the expression of the *Dio2* gene by CRISPR/Cas9 technology in mouse C2C12 muscular cells (see Materials and Methods). C2C12 cells were also stably transfected with a control CRISPR/Cas9 plasmid and used as control cells (CTR) in all the experiments. We selected two different clones with effective mutation of the *Dio2* gene (D2 KO cells, Appendix A). Intracellular T3/T4 ratio reduced in D2 KO cells compared to CTR cells, thus confirming the reduction on T4-to-T3 conversion in the D2 mutated clones (Figure 1A).

Myoblast proliferation was enhanced in D2-KO versus CTR cells, in agreement with the concept that D2 inactivation increases cell proliferation (Figure 1B,C) [15,16]. To assess if enhanced cell proliferation was due to reduction of TH transactivation activity, we measured the expression levels of three TH target genes (p21, p27 and Cyclin-D1) [17,18]. In comparison with CTR cells, mRNA expression of Cyclin-D1 was enhanced, while p21 and p27 expression reduced in D2 KO versus CTR cells (Figure 1D), confirming that D2 silencing reduces the expression of positive TH target genes (p21 and p27) and increases the expression of a negative TH target gene (Cyclin-D1).

Cyclin-D1 protein levels were also robustly increased in asynchronized and synchronized D2 KO cells compared to CTR cells (Figure 1E). Consistently, expression of p21 and p27 protein levels was reduced in D2-depleted cells (Figure 1E). Moreover, analysis of the cell cycle progression showed that loss of D2 results in a higher percentage of duplicating cells when compared to CTR cells (Figure 1F). To further confirm that the TH reduces cell proliferation by a genomic mechanism, we transfected C2C12 cells with a TRPV plasmid, coding for a mutated form of TRα that acts as a dominant negative isoform on the endogenous TRs [19]. Furthermore, we generated two different plasmids encoding mutated TRα (TRα-D mut) and TRβ (TRβ-D mut) isoforms incapable of nuclear translocation since the nuclear translocation sequence in the D Domain of TRα and TRβ proteins were mutated as described in the Materials and Methods (Appendix A). TRPV and TR-D mut-transfected C2C12 cells were characterized by reduced expression of p21 and p27 and by up-regulation of Cyclin-D1 mRNA (Appendix A).

These data confirm that D2 is a critical inducer of the nuclear action of TH and that loss of D2 and inhibition of TRs enhance cell proliferation of myoblasts by increasing the expression of proliferative genes and reducing the expression of genes involved in cell cycle arrest.

### 2.2. Silencing of β3-Integrin Inhibits Cell Proliferation of Muscle Cells

We next evaluated the effects of αVβ3-Integrin inactivation on C2C12 cell proliferation. CRISPR/Cas9 technology was used to generate two different clones with stable inactivation of the *β3-Integrin* isoform (β3-Integrin, see Materials and Methods) (Appendix A). Intracellular T3/T4 ratio was unchanged when compared to CTR cells, thus confirming that inhibition of β3-Integrin does not modify the intracellular T3 and T4 levels (Figure 2A).

Muscle cell proliferation was attenuated in β3-Integrin KO cells when compared to CTR cells, in agreement with previous reports indicating that binding of T4 to αVβ3-Integrin fosters cell proliferation (Figure 2B,C) [20,21]. As a marker of T4-Integrin interaction, we measured the levels of phospho-ERK 1/2 that were reduced in β3-Integrin KO versus CTR cells, while levels of the anti-proliferative phospho-p38 MAPK were enhanced in β3-Integrin KO versus CTR cells (Figure 2D). Moreover, cell cycle analysis confirmed that interfering with the αVβ3-Integrin action reduced the proportion of proliferating cells (Figure 2D). To further confirm that loss of T4-Integrin interaction inhibits muscle cell proliferation, we treated C2C12 cells with two inhibitors of the αVβ3-Integrin action (TETRAC and RGD peptide). Also in this case, Integrin signaling inhibition resulted in decreased levels of phospho-ERK 1/2, enhanced levels of anti-proliferative phospho-p38 MAPK (Appendix A) and reduced cell proliferation (Appendix A). Notably, we observed that inhibition of D2 expression resulted in elevated levels of phospho-ERK 1/2 expression and reduced expression of phospho-p38 MAPK, which are respectively down- and up-modulated in β3-Integrin KO cells (Appendix A). Accordingly, inhibition of αVβ3-Integrin led to reduced levels of Cyclin-D1 and increased levels of p21 (Appendix A). These data suggest that ablation of the T4-to-T3 conversion in D2 KO cells, by elevating the availability of T4 (Figure 1A), enhances the effects of αVβ3-Integrin signal, increasing the proliferation rate and the levels of phospho-ERK 1/2. Similarly, inhibition of the αVβ3-Integrin signal potentiates the T3-dependent reduction of cell proliferation rate.

### 2.3. Muscle Cell Differentiation Is Positively Regulated by Both Genomic and Non-Genomic TH Action

Next, C2C12 D2 KO, β3-Integrin KO and CTR cells were cultured in a differentiation medium in order to assess the contribution of the two different pathways in myoblast differentiation. Since D2 and D2-produced T3 are key mediators of myogenic differentiation, as expected, D2 inactivation caused reduced muscle cell differentiation. Indeed, the expression of the myogenic markers MyoD and Myogenin was down-regulated in D2 KO cells compared to CTR cells (Figure 3A–C). Furthermore, the TH negative target gene MHC-I was up-regulated, while MHC-II was down-regulated in D2 KO compared to CTR cells, thus confirming the attenuation of the genomic action of TH (Figure 3D–F). Notably, β-Integrin cells were also characterized by reduced cell differentiation ability, with drastic reduction of the myogenic markers Myogenin (Figure 3H,I) and MHC-II (Figure 3K,L). However, no difference in MyoD expression was observed (Figure 3G,I) nor in the inhibition of MHC-I expression (Figure 3J), both effects of direct, genomic action of TH. Consistently with the mRNA expression analysis, immunofluorescence analysis demonstrated drastic reduction of the slow, oxidative fibers (type I fibers, MHC I, here referred to as MF20) in D2- and β3-Integrin-depleted cells compared to CTR cells (Figure 3M).

These data show that D2-dependent and Integrin-dependent TH actions both converge in induction of myoblast differentiation.

### 2.4. Both Genomic and Non-Genomic Actions of TH Reduce Redox State of Muscle Cells

To evaluate the contribution of the two signaling pathways on the redox state of muscle cells, we measured the ROS production by FACS analysis. At basal level, ROS production was significantly enhanced in muscle cells by both D2 and Integrin inhibition (Figure 4A). Importantly, the expression of a set of antioxidant genes (*GCLC, GPX, HO1* and *NFE2L2*) was down-regulated in both D2 KO and β3-Integrin KO cells, while the mitochondrial scavenger SOD2 was specifically reduced at transcriptional level in D2 KO cells (Figure 4B). Treatment of cells with exogenous T3 rescued the enhanced ROS production observed in D2 KO and β3-Integrin KO cells, while T4 reduced ROS production observed in β-Integrin cells but, as expected, had no effects in D2 KO cells (Figure 4C). Moreover, TH treatment rescued the expression of antioxidant genes, *SOD2, GCLC, GPX, HO1* and *NFE2L2* (Figure 4D–H), in accordance with the protective role of TH activation against basal oxidative stress [14].

The reduction of scavenger enzymes in both D2 KO and β3-Integrin KO cell lines compared to CTR cells confirmed that TH reduces ROS production by potentiating the scavenger role of antioxidant genes.

## 3. Discussion

Besides the canonical, nuclear action of THs, extranuclear mechanisms of TH action have been widely reported in a series of cellular contexts [22]; effects can be either initiated by membrane binding of THs with receptor proteins or by intracellular binding with cytoplasmic partners. In both cases, non-genomic actions are defined as mechanisms that are not mediated by interaction of THs with nuclear receptors and that are not blocked by inhibitors of transcription and translation [3,5,23]. Among the specific actions of T3 and T4 in the non-genomic mechanisms, T4 is mostly implicated in the binding with αVβ3-Integrin and the activation of the MAPK signaling pathway, while T3 is the principal activator of the Akt/PI3K pathway [24].

The aim of the present work was to dissect the differential contribution of the genomic and non-genomic actions of THs in muscle cells. As a paradigm of genomic action of THs we investigated the mechanism of T4-to-T3 conversion operated by the type 2 deiodinase, which is an essential step for TH activation and migration to the nucleus. Moreover, we analyzed the effects of TH mediated by the T3-TR binding in muscle cells as an additional indicator of the genomic action of THs. Conversely, binding of T4 with αVβ3-Integrin was used as a model of non-genomic action of THs (Figure 5).

Inhibition of the *Dio2* gene expression by CRISPR/Cas9 technology led to reduced T4-to-T3 conversion in muscle cells, thus representing an ideal model for attenuation of nuclear T3 production and the consequent genomic mechanism. When interfered with for D2, muscle cells showed an enhanced proliferation rate and reduced cell differentiation ability. These results confirm the previous findings that the D2-mediated TH activation fosters muscle cell differentiation while attenuating myotube formation [15,25,26,27]. Similarly, in different tissue compartments, TH acts as a pro-differentiation agent [28,29]. Loss of αVβ3-Integrin expression caused a drastic reduction of both myoblast proliferation and differentiation ability, thus showing that THs are potent inducers of muscle cell differentiation acting via both genomic and non-genomic mechanisms [1,15], confirming the previous observation that the αVβ3-Integrin signaling induces satellite cell differentiation by the activation of Rac1 and focal adhesion formation [30]. These results are in agreement with the thyroid myopathy found in patients with both hypo- and hyperthyroidism [1]. To further assess how the two signaling pathways are interconnected, we evaluated the effect of D2 ablation on non-genomic targets such as ERK 1/2 and p38-MAPK, observing that these markers are inversely regulated in D2 KO cells when compared with β3-Integrin KO cells, thus suggesting that inhibition of the genomic signal results in activation of the non-genomic pathway (probably due to the increased levels of T4 in D2 KO cells). Consistently, inhibition of the integrin signal resulted in reduction of Cyclin-D1 and elevation of p21 expression, confirming that interfering with the non-genomic signal potentiates the genomic signal for the control of the cell proliferation rate of muscle cells.

The ability of THs to regulate many different metabolic pathways in skeletal muscle has been known for over a century [31]. THs affect muscle cell metabolism not only by regulating expression of metabolic enzymes but also by acting as major determinants of muscle fiber isoform shift and promoting the conversion of red slow (type I) fibers to white fast (type II) fibers [32]. As a consequence, hyperthyroid conditions are characterized by a higher percentage of white fibers, with the opposite occurring in hypothyroid conditions. Moreover, THs induce the transcription of the Ca^2+^ pump SERCA and foster muscle contraction, along with the induction of the Na^+^/K^+^ ATPase, GLUT4 and PGC1-α, representing critical metabolic regulation of TH in muscle fibers [12,33].

A key metabolic action of THs that is still poorly understood is the regulation of the redox state. Indeed, while THs are powerful inducers of the basal metabolic rate and cellular respiration, thus resulting in augmented production of intracellular ROS [13], at the same time, THs enhance the transcription of different ROS scavenger enzymes [14]. In line with this concept, we observed that inhibition of both D2 and αVβ3-Integrin pathways led to global increased ROS production in muscle cells. Indeed, our data showed that the expression levels of a panel of antioxidant genes are reduced when both genomic and non-genomic actions of THs are inhibited. Thus, the induction of antioxidant genes might be viewed as an adaptive mechanism to balance the redox homeostasis secondary to TH-induced oxidative stress. Thus, recent research points to the role of THs as critical determinants in maintaining the homeostatic control of ROS dynamics while enhancing cell respiration rate.

Our data demonstrate a convergent role of genomic and non-genomic action of TH in myogenic differentiation induction. Thus, the observed reduction of cell differentiation and increased basal ROS levels confirm that the overall effect of TH signal inhibition is coupled to reduction in myoblast differentiation partially due to enhanced ROS production. This is in line with the concept that, while a balanced equilibrium on ROS production and scavenger activity of antioxidant genes are essential to ensure a correct progression from cell proliferation to differentiation during myogenesis, enhanced ROS production is often the leading cause of inhibition of myogenic differentiation [34].

Keeping in mind that the main hormone produced by the thyroid gland is T4, which thus represents the major systemic supply of THs to the target cells, one could argue that genomic and non-genomic mechanisms might compete for the same substrate T4 for their actions. However, our data show that, while this is true for the control of cell proliferation, which is inversely regulated by the genomic and non-genomic action of TH, in other cases, such as the control of myogenic differentiation, genomic and non-genomic mechanisms lead to similar final effects, although mediated by different intracellular effectors (Figure 5).

In conclusion, our work provides interesting evidence that many effects exerted by TH in muscle cells are commonly modulated by genomic mechanisms, via the nuclear TR receptors, and also by non-genomic actions, thus confirming the existence of cooperative crosstalk between genomic and non-genomic pathways of TH [20].

## 4. Materials and Methods

### 4.1. Cell Cultures

C2C12 cells were obtained from ATCC and cultured in Dulbecco’s Modified Eagle Medium (Gibco, Thermo Fisher Scientific, Waltham, MA, USA) supplemented with 10% Fetal Bovine Serum (Gibco), 1% L-Glutamine (Gibco) and 1% Penicillin/Streptomycin (Gibco) (proliferating medium). For studies in proliferative conditions, C2C12 cells were grown at 40–50% confluence. For experiments in differentiation conditions, cells were grown at 60–70% confluence and then induced to differentiate in Differentiation Medium, DMEM with 2% Horse Serum (cod. H1138, Sigma-Aldrich, St. Louis, MO, USA), Insulin 10 μg/mL (cod. I2643, Sigma-Aldrich) and Transferrin 5 μg/mL (cod. T8158, Sigma-Aldrich). In some experiments, THs, T3 (cod. T6397—T3 (3,3′,5-Triiodo-L-thyronine sodium salt), Sigma-Aldrich St. Louis, Missouri, USA) and T4 (cod. T2501—T4 (L-Thyroxine sodium salt pentahydrate), Sigma-Aldrich St. Louis, Missouri, USA) were added in culture medium at a 30.0 nM final concentration. In other experiments, Tetrac (3,3′,5,5′-Tetraiodothyroacetic acid, cod. T3787, Sigma-Aldrich) and RGD peptide (Arg-Gly-Asp ≥ 97% TLC, cod. A8052, Sigma-Aldrich) were added in culture medium at a 40.0 μM final concentration. 

### 4.2. Constructs and Transfections

All transient transfection experiments were performed using Lipofectamine 2000 (Invitrogen™, Carlsbad, CA, USA) according to the manufacturer’s instructions. The set of vectors for thyroid hormone mutant receptor expression and the CMV FLAG control plasmids was purchased from Clontech Laboratories (Mountain View, CA, USA). Direct sequencing was carried out to verify the correct sequence (Eurofins Genomics). TRPV mutant plasmid was kindly provided by Dr. Sheue-yann Cheng. The TRPV mutant carries a mutation in the mouse TRα gene as described by Kaneshige, M. et al. [19] that leads to a frameshift mutation of the TRα gene. For the generation of TRa and TRb D-mut plasmids, PCR site-directed mutagenesis was performed by using a version of the high-fidelity DNA polymerase, i.e., Q5 DNA polymerase (New England BioLabs, Massachusetts, USA) in order to ensure an accurate amplification. The PCR product carrying the mutation of interest was cleaved with EcoR1, BglII and PvUII restriction enzymes (New England BioLabs) and inserted in place of the wild-type sequence into TRα and the TRβ plasmids [35] (digested with EcoR1, BglII and PvUII restriction enzymes). FLAG tag was also fused to the amino termini of the two mutants to obtain FLAG-TRα and FLAG-TRβ D-mut plasmids. FLAG-TRa D-mut and TRb D-mut were constructed as also described by Zhu, X.G. et al. [36] by replacing the sequence A^1357^-A^1358^-C^1360^-G^1361^-A^1369^-A^1370^-C^1372^-G^1373^-A^1375^-A^1376^ (Lys^453^-Arg^454^-Lys^457^-Arg^458^-Lys^459^) with G^1357^-C^1358^-G^1360^-C^1361^-G^1369^-C^1370^-G^1372^-C^1373^-G^1375^-C^1376^ (Ala^453^-Ala^454^-Ala^457^-Ala^458^-Ala^459^) and A^1522^-A^1523^-A^1525^-G^1526^-A^1534^-A^1535^-C^1537^-G^1538^-A^1540^-A^1541^ (Lys^1507^-Arg^1508^-Lys^1511^-Arg^1512^-Lys^1513^) with G^1522^-C^1523^-G^1525^-C^1526^-G^1534^-C^1535^-G^1537^-C^1538^-G^1540^-C^1541^ (Ala^1507^-Ala^1508^-Ala^1511^-Ala^1512^-Ala^1513^), respectively. TRa D-mut and TRb D-mut mutagenesis was then confirmed by DNA sequencing and Nucleus–Cytosol protein fractioning.

### 4.3. Dio2 and 3-Integrin Targeted Mutagenesis

Targeted mutagenesis of *Dio2* in C2C12 cells was achieved using the DIO2 Plasmid KO CRISPR/Cas9 (m) system from Santa Cruz Biotechnology (cod. sc-420003, Dallas, Texas, USA). Targeted mutagenesis of *β3-Integrin* in C2C12 cells was achieved using the *β3-Integrin* CRISPR/Cas9 KO Plasmid (m) system from Santa Cruz Biotechnology (cod. sc-421175). C2C12 control cells were stably transfected with the Control CRISPR/Cas9 plasmid (cod. sc-418922, Santa Cruz Biotechnology, Dallas, TX, USA). Twenty-four hours after transfection, the cells were sorted using fluorescence-activated cell sorting (FACS) for green fluorescent protein (GFP) expression. Single clones were analyzed by PCR to identify alterations in coding regions. *Dio2* exon 1 (clone 2 and clone 8) was sequenced to identify the inserted mutations. Analogously, *β3-Integrin* exon 3 (clone 9) and exon 4 (clone 3) were sequenced to identify the inserted mutations. All the experiments in D2 KO and β3-Integrin KO cells were repeated in two different clones to avoid off-target effects.

### 4.4. Western Blot Analysis

Total protein extracts from cells were run on a 10% or 15% SDS-PAGE gel and transferred onto an Immobilon-P transfer membrane (Millipore, Burlington, MA, USA). The membrane was then blocked with 5% non-fat dry milk (Bio-Rad, Hercules, CA, USA) in PBS-0.2%Tween, probed with primary antibodies (indicated in Appendix A) overnight at 4 °C, washed and incubated with horseradish peroxidase-conjugated anti-mouse immunoglobulin G secondary antibody (cod. 1706516, Bio-Rad) or anti-rabbit immunoglobulin G secondary antibody (cod. 1706515, Bio-Rad). Anti-Tubulin antibody was used as loading control. Band detection was performed using an ECL kit (Millipore, Burlington, MA, USA). The gel images were analyzed using ImageJ software (NIH Image, Bethesda, MD, USA), and all Western blots were run in triplicate.

For subcellular fractionation, cells were lysed in a buffer containing 10 mM Tris HCl pH 7.9, 10 mM KCl, 1.5 mM MgCl_2_ and 1.5 mM DTT with protease inhibitors and carefully homogenized by pipetting several times. Then the lysate was passed through a 25 Ga needle 10 times using a 1 mL syringe and left for 30 min at 4 °C. Nuclei were precipitated by centrifugation at 6000× *g* for 10 min at 4 °C, and the cytoplasmic-enriched fraction (C) was removed. For preparation of nuclear-enriched (N) fractions, the crude nuclear pellets were lysed in a buffer composed of 50 mM Tris pH 7.5, 0.3 M sucrose, 0.42 M KCl, 5 mM MgCl, 0.1 mM EDTA and 2 mM DTT with protease inhibitors and resuspended by pipetting. After 20 min at 4°C, the lysate was centrifuged at 13000 rpm for 10 min at 4 °C. The supernatant containing the nuclear-enriched (N) fraction was then removed. 

### 4.5. Real-Time PCR

Messenger RNAs were extracted with TRIzol reagent (Life Technologies Ltd., Carlsbad, CA, USA). Complementary DNAs were prepared with 5X All-In-One RT Master Mix (Applied Biological Materials, abm, Heidelberg, Germany) as indicated by the manufacturer. The cDNAs were amplified by real-time PCR in a CFX Connect Real-Time PCR Detection System (Bio-Rad) with the fluorescent double-stranded DNA-binding dye SYBR Green (Bio-Rad). Specific primers for each gene were designed to work under the same cycling conditions (95 °C for 3 min followed by 50 cycles at 95 °C for 30 s and 60 °C for 1 min), thereby generating products of comparable sizes (about 200 bp for each amplification). Primer combinations were positioned whenever possible to span an exon–exon junction and the RNA digested with DNAse to avoid genomic DNA interference. Primer sequences are indicated in Appendix A. For each reaction, standard curves for reference genes were constructed based on six four-fold serial dilutions of cDNA. All samples were run in triplicate. The template concentration was calculated from the cycle number when the amount of real-time PCR product passed a threshold established in the exponential phase of the real-time PCR. The relative amounts of gene expression were calculated with cyclophilin A expression as an internal standard (calibrator). The results, expressed as N-fold differences in target gene expression, were determined as follows: *N* * target = 2 ^(ΔCt sample−ΔCt calibrator)^.

### 4.6. Immunofluorescence Analysis

For immunofluorescence staining, C2C12 cells were fixed for 20 min with 30%-Acetone/70%-Methanol and permeabilized in 0.1% Triton X-100 (cod. T8787, Sigma-Aldrich) then blocked with 0.2% BSA/PBS and washed in PBS. Cells were then incubated with Myosin heavy-chain (MHC) primary antibody overnight at 4 °C. Donkey anti-Mouse IgG (H + L) highly cross-adsorbed secondary antibody and Alexa Fluor 594 secondary antibody (cod. A21203, Invitrogen ™, Carlsbad, CA, USA) incubation was carried out at room temperature for 1 h, followed by washing in 0.2% Tween/PBS. Images are the representation of experiments performed in triplicate and were acquired with a Leica DMi8 microscope and the Leica Application Suite LAS X Imaging Software (Leica Microsystems, Wetzlar, Germany).

### 4.7. Colony Formation Assay

To evaluate colony formation, 3.000 D2 KO, β3-Integrin KO and CTR cells were seeded in cell culture plates to form colonies. One week after plating, cells were washed with PBS and stained with 1% Crystal Violet (cod. C6158, Sigma Aldrich) in 20% ethanol for 30 minutes at room temperature. Cells were washed twice with PBS, and colonies were counted (*n* = 4).

### 4.8. MTT Assay

The viability of D2 KO, β3-Integrin KO and CTR cells was determined using a standard MTT (3-(4, 5-dimethylthiazol-2-yl)-2,5-diphenyltetrazolium bromide) assay. All the treatments were done using 1 × 10^3^ cells/well in 96-well plates (*n* = 4). The purple formazan crystals were dissolved in DMSO (100 μL/well, cod. D2650, Sigma-Aldrich), and the absorbance was recorded on a microplate reader (VICTOR^®^ High Performance Multimode Detection Technologies) at a wavelength of 570 nm.

### 4.9. HPLC-MS Measurement of T3 and T4

Standard stock solutions of all target analytes (3,3′,5,5′-Tetraiodo-L-thyronine (L-thyroxine (T4) and 3,3′,5-triiodothyroxine (T3)) were prepared in methanol. Dilutions of each standard were prepared in methanol/water (*v*/*v*, 50/50). Ten milliliters of cell media were deproteinated using 9 volumes of cold acetone and then centrifuged at 14.000 rpm. The supernatants were reduced to 200 μL under N_2_ for instrumental analysis. The HPLC system Jasco Extrema LC-4000 system (Jasco Inc., Ithaca, NY, USA) was coupled to an Advion Expression mass spectrometer (Advion Inc., Ithaca, NY, USA) equipped with an ESI (electrospray ionization) source. Ten millimoles of ammonia acetate in deionized water were used as the aqueous mobile phase, and 0.1% acetic acid in methanol was used as the organic mobile phase. The analyses were performed in the positive ESI mode. Six replicates were run for each sample.

### 4.10. Measurement of Cellular ROS

Total ROS levels were measured using the 5-(and-6)-chloromethyl-2,7-dichlorodihydrofluorescein diacetate CM-H_2_DCFDA probe (cod. D399, Invitrogen ™), according to the manufacturer’s instructions. Briefly, after trypsinization, cells were collected and rinsed with PBS. Cells were then resuspended and incubated in pre-warmed PBS containing 5.0 μM CM-H_2_DCFDA in the dark for 30 min at 37 °C. Intracellular fluorescence was then quantified using a FACS Canto2 (Becton, Dickinson and Company, CA, USA), (*n* = 3).

### 4.11. Cell Cycle Analysis

D2 KO, β3-Integrin KO and CTR C2C12 cells were first synchronized after 24 h of serum starvation. Cells were fixed in ice-cold 70% ethanol at −20 °C. At least 10.000 cells were analyzed by FACS (FACS Canto2, Becton, Dickinson and Company) after staining with 5.0 μg/mL propidium iodide (PI) and exposure to 0.25 mg/mL RNase I (Sigma-Aldrich). Data were analyzed with the MODFIT Lt3.0 Software. For carboxyfluorescein succinimidyl ester (CFSE) in vivo labeling, the CFSE labeling solution (1.0 μM) was added to 1.0 mL cell suspensions in PBS and incubated for 10 min at room temperature, following which cells were washed with regular medium to quench any free dye in solution. Cells were then plated in regular growth medium for different time points and harvested for cell sorting using a FACS Aria system (BD) (*n* = 3).

## 5. Statistics

The results are reported as means ± SD throughout the study. Differences between samples were assessed by the Student’s two-tailed t test and ANOVA for independent samples. Relative mRNA levels (in which the control sample was arbitrarily set as 1) are reported as results of real-time PCR in which the expression of cyclophilin A served as a housekeeping gene. In all experiments, differences were considered significant when *p* was less than 0.05. Asterisks indicate significance at * *p* < 0.05, ** *p* < 0.01 and *** *p* < 0.001 throughout.

## Figures and Tables

**Figure 1 ijms-22-07175-f001:**
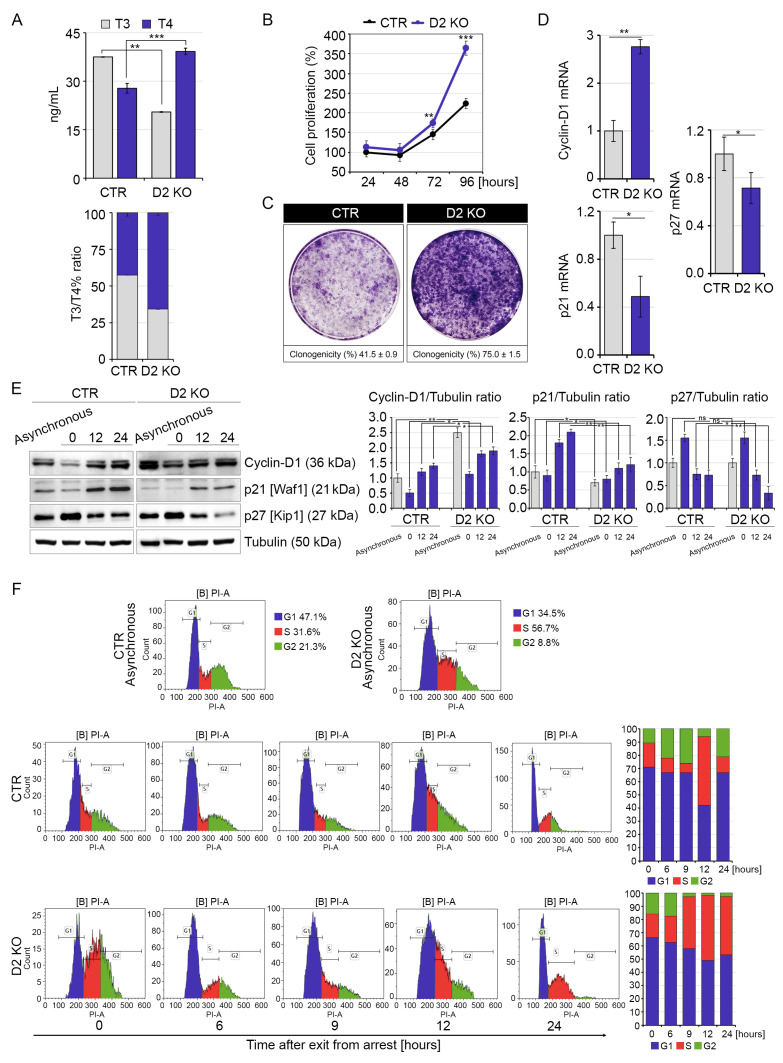
D2 knock-out promotes cell division rate and progression through the cell cycle compared to CTR cells. (**A**) Levels of T3 and T4 measured by HPLC-MS in the culture medium of D2 KO and CTR cells (top). The same levels are expressed as T3/T4% ratio (bottom). (**B**) MTT assays were used to determine the rate of cell proliferation of D2 KO and CTR cells at 24, 48, 72 and 96 h after seeding cells. (**C**) Clonogenicity of D2 KO and CTR cells was determined by seeding cells (3000 per well in 6-well plates) and growing for 1 week. (**D**) Cyclin-D1, p21 and p27 expression was evaluated by real-time PCR in D2 KO and CTR cells grown in proliferative conditions (48 h following cell seeding). (**E**) Western blot analysis of Cyclin-D1, p21 and p27 in asynchronized and synchronized D2 KO and CTR cells after cell cycle re-initiation at 0, 6, 12 and 24 h after release from serum starvation. Tubulin expression was measured as a loading control. Quantification of the protein levels of Cyclin-D1, p21 and p27 versus Tubulin levels is represented by histograms. Data represent the mean of three independent experiments. (**F**) Cell cycle distribution was measured in asynchronized and synchronized D2 KO and CTR cells at 0, 6, 12 and 24 h after release from serum starvation. Cells were analyzed with flow cytometry after propidium iodide staining (*n* = 3). * *p* < 0.05, ** *p* < 0.01, *** *p* < 0.001.

**Figure 2 ijms-22-07175-f002:**
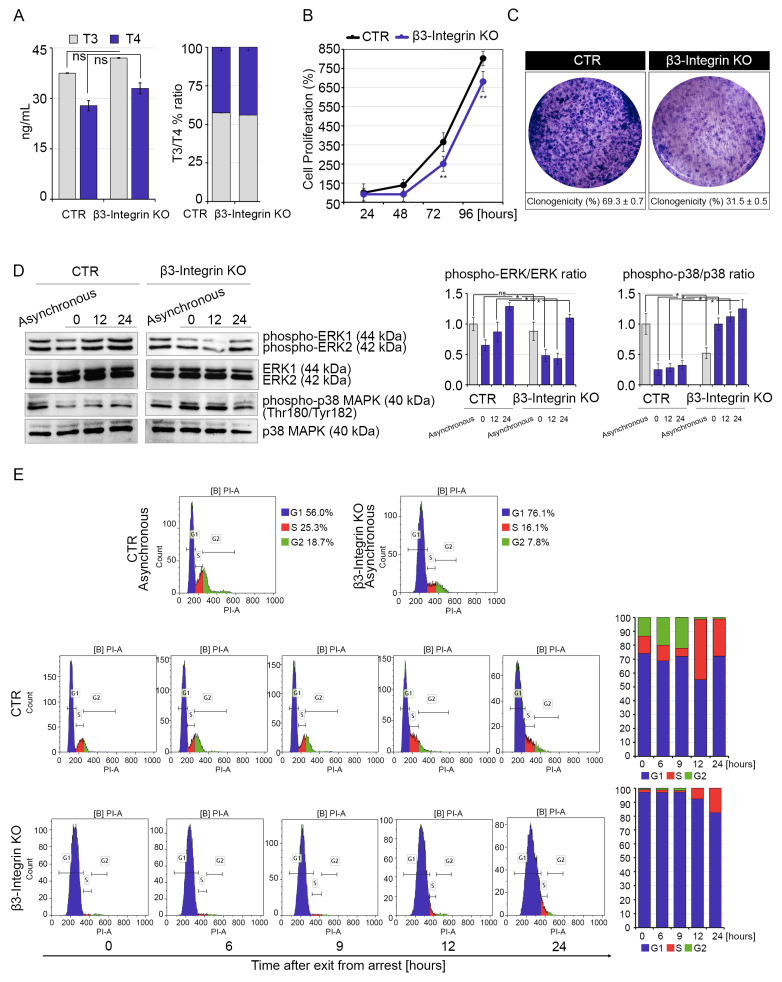
β3-Integrin knock-out impairs cell division rate and progression through the cell cycle compared to CTR cells. (**A**) Levels of T3 and T4 measured by HPLC-MS in the culture medium of β3-Integrin KO and CTR cells (left). The same levels are expressed as T3/T4% ratio (right). (**B**) MTT assays were used to determine the rate of cell proliferation of β3-Integrin KO and CTR cells at 24, 48, 72 and 96 h after seeding cells. * *p* < 0.05, ** *p* < 0.01. (**C**) Clonogenicity of β3-Integrin KO and CTR cells was determined by seeding cells (3000 per well in 6-well plates) and growing for 1 week. (**D**) Western blot analysis of phospho-ERK 1/2 and phospho-p38 MAPK in asynchronized and synchronized β3-Integrin KO and CTR cells after cell cycle re-initiation at 0, 6, 12 and 24 h after release from serum starvation. ERK 1/2 and p38 MAPK expression was measured as a loading control. Quantification of the protein levels of phospho-ERK 1/2 versus ERK 1/2 and phospho-p38 MAPK versus p38 MAPK levels are represented by histograms. Data represent the mean of three independent experiments. (**E**) Cell cycle distribution was measured in asynchronized and synchronized D2 KO and CTR cells at 0, 6, 12 and 24 h after release from serum starvation. Cells were analyzed with flow cytometry after propidium iodide staining (*n* = 3). ns = non-significant, ** *p* < 0.01.

**Figure 3 ijms-22-07175-f003:**
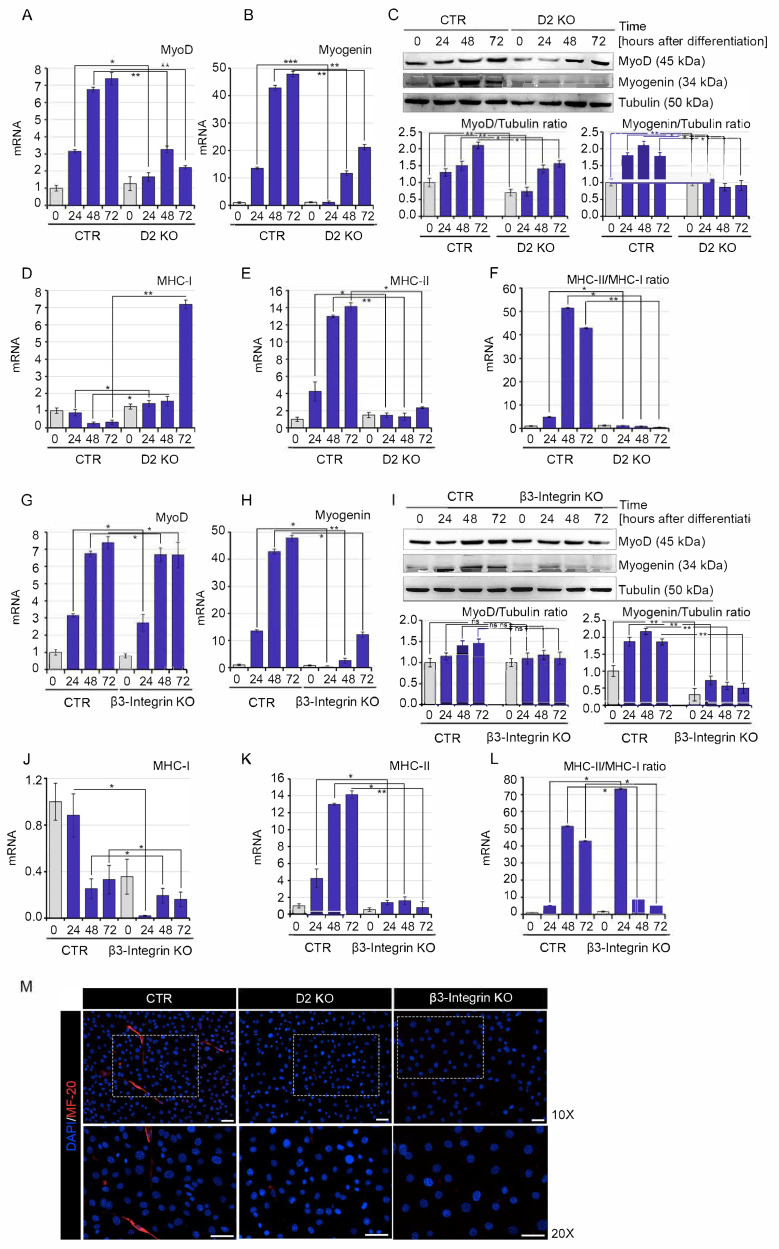
D2 and β3-Integrin knock-out block TH-mediated myogenic differentiation. (**A**,**B**) MyoD and Myogenin expression was evaluated by real-time PCR in D2 KO and CTR cells grown in differentiative conditions at 0, 24, 48 and 72 h after differentiation. (**C**) Western blot analysis of MyoD and Myogenin in the same cells as in A, B. Tubulin expression was measured as a loading control. Quantification of the protein levels of MyoD and Myogenin versus Tubulin levels is represented by histograms. Data represent the mean of three independent experiments. (**D**,**E**) Expression levels of the myosin heavy-chain isoforms (MHC-I and MHC-II) were measured by real-time PCR in the same cells as in A, B. MHC-II/MHC-I ratio was shown in (**F**). (**G**,**H**) MyoD and Myogenin expression was evaluated by real-time PCR in β3-Integrin KO and CTR cells grown in differentiative conditions at 0, 24, 48 and 72 h after differentiation. (**I**) Western blot analysis of MyoD and Myogenin in the same cells as in G, H. Tubulin expression was measured as a loading control. Quantification of the protein levels of MyoD and Myogenin versus Tubulin levels is represented by histograms. Data represent the mean of three independent experiments. (**J**,**K**) Expression levels of the myosin heavy-chain isoforms (MHC-I and MHC-II) were measured by real-time PCR in the same cells as in G, H. MHC-II/MHC-I ratio is shown in (**L**). (**M**) Immunofluorescence analysis of MF20 (MHC) expression was performed in differentiated D2 KO, β3-Integrin KO and CTR cells. Magnification 10X and 20X. Scale bars represent 50 μm. ns = non-significant, * *p* < 0.05, ** *p* < 0.01, *** *p* < 0.001.

**Figure 4 ijms-22-07175-f004:**
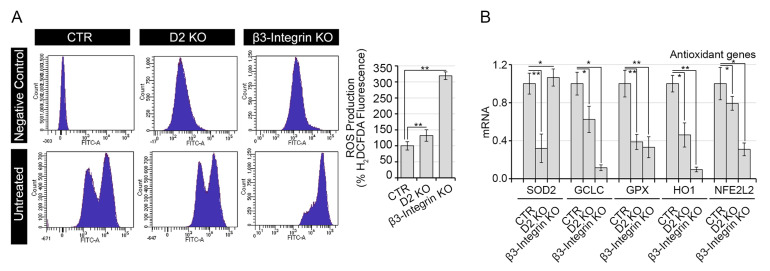
D2 and β3-Integrin KO reduce endogenous ROS production. (**A**) Basal ROS were measured by FACS analysis in D2 KO, β3-Integrin KO and CTR cells. Right panel shows the relative % mean of H_2_DCFDA fluorescence compared to CTR cells arbitrarily set as 100%. (**B**) mRNA expression of a panel of antioxidant genes, as SOD2, GCLC, GPX, HO1 and NFE2L2, was evaluated by real-time PCR in D2 KO, β3-Integrin KO and CTR cells. (**C**) Basal ROS were measured by FACS analysis in D2 KO, β3-Integrin KO and CTR cells treated with 30.0 nM T3 or T4. Right panels show the relative % mean of H_2_DCFDA fluorescence of D2 KO, β3-Integrin KO and CTR cells, each compared to its untreated control. (**D**–**H**) mRNA expression of a panel of antioxidant genes, as SOD2, GCLC, GPX, HO1 and NFE2L2, was evaluated by real-time PCR in D2 KO, β3-Integrin KO and CTR cells treated with 30.0 nM T3 or T4. * *p* < 0.05, ** *p* < 0.01, *** *p* < 0.001.

**Figure 5 ijms-22-07175-f005:**
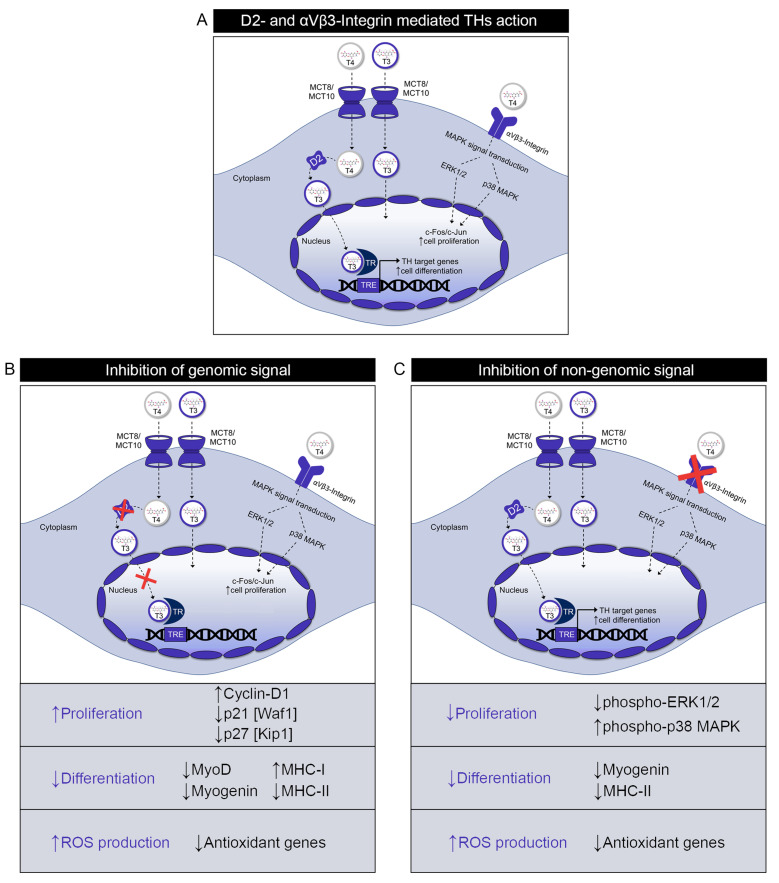
Thyroid hormones act via both genomic and non-genomic mechanisms to regulate muscle physiology. (**A**) Schematic representation of D2-mediated TH activation and αVβ3-Integrin dependent action of THs. (**B**) Consequences of inhibition of D2 expression and the relative reduction of TH activation in D2 KO muscle cells. (**C**) Effects of inhibition of αVβ3-Integrin expression in muscle cells. The purple arrows indicate the up-regulation and the down-regulation of the processes influenced by D2 KO and β3-Integrin KO, while the black arrows were used to specify the involved genes.

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
