# Peer review of "Selective Inhibition of Genomic and Non-Genomic Effects of Thyroid Hormone Regulates Muscle Cell Differentiation and Metabolic Behavior"

_ijms, 2021, doi:10.3390/ijms22137175_

Round 1

Reviewer 1 Report

Nappi et al investigate the contribution of genomic and non-genomic TH signaling on cell proliferation and differentiation and ROS production in muscle cells (C2C12). They address the genomic TH action via deletion of Dio2 as mediator for T4 conversion into T3, thereby reducing T3 that is to some extent responsible for the TR/genomic pathway, and, they address the non-genomic pathway via deletion of αVβ3-Integrin, which eliminates T4 action via αVβ3, i.e. activation of the ERK1/2 signaling pathway. They propose that the non-genomic pathway is responsible for myoblast proliferation, that both (genomic and non-genomic) are triggers for cell differentiation and regulators of the redox signaling of the cells.

The study addresses differential functions of TH, which is of general interest. However, the study lacks several mechanistic links, conclusions drawn need further substantiation and a thorough discussion of the potential impact of their study on biological, pathological or physiological processes is needed.

Specific comments:

  • Inhibition of non-genomic signaling by integrin-KO inhibits cell proliferation via ERK1/2 and p38 signaling. How does the integrin-KO influence cyclin D1, p21 and p27 signaling and, vice versa, how does D2-KO impact on ERK1/2 and p38 signaling? These questions should be addressed to validate the proposed signaling pathway/interconnection of T4/T3 and integrin-mediated cell proliferation.
  • The authors included data on ROS production and mRNA levels of several “antioxidant” genes. How is ROS or are these genes related to cell proliferation and myoblast differentiation? This link is not validated experimentally and is not convincingly discussed.
  • To support the conclusion that will be drawn by the readers that “non-genomic signaling” is responsible for cell proliferation, the authors also use several mutants, that are supposed to act dominant negative on genomic signaling, and analyse their effects on the mRNA levels of cyclin D1, p21 and p27. The “background” of the mutants should be included to better understand their action in the context of the experimental design. Why should a gene construct with an NLS sequence localize to the cytosol? It is a nuclear localization sequence?
  • Which signaling pathway is involved in myoblast differentiation? That Dio2 is involved in cell differentiation has already been reported (Bloise et al. 2018). ERK/p38? Which signaling events are thought to promote myoblast differentiation? Or does integrin KO interfere in cell differentiation independently of T4? Is there a contradiction in figure M to figure D/F where they observe an increase in MHC expression? That increase is not seen in immunofluorescence staining.
  • 1: The authors propose that D2-KO promotes cell proliferation of C2C12 myoblast cells. They evaluate proliferation using the MTT assay and analysed TH target gene expression of cyclin D1, p21 and p27 on mRNA and protein level. Indeed, the proliferation rate was increased 72 and 96 hours after seeding and the mRNA data show convincing differences between ctr and KO cells. The data on protein expression, however, are somewhat confusing. If ctr and KO cells are thought to be compared, the samples need to be run on the same gel/blot: If the authors are saying cyclin D1 is increased or p21/p27 decreased, what do they mean? The basal value? The induction over time after seeding? Why were different time points chosen than in the proliferation assay? The increase over time in control cells seems significantly higher than in the D2-KO cells and similar for p21 and p27… The quantification of the blots does not indicate significances and it is unclear how normalization was done. The protein data need more thorough analyses and explanations.
  • Similarly, as described above for Fig. 1, it is unclear in Fig. 2 what the authors mean by phospho-ERK1/2 is reduced by integrin-KO. Which time point is compared, the increase, the different cell lines (if yes – new Western blots need to be done to be able to compare the data), how was the analysis done? Statistics in Fig. 2D is missing. At time point 24h, phospho-ERK1/2 does not seem to be reduced in graph… phospho-p38 signaling was also analysed since it is thought to be an anti-proliferative pathway. Here, again, it is completely unclear what the authors mean by “increased”. How does integrin-signaling impact on ERK and p38 signaling? What is known, what is the hypothesis? Does T3 contribute to p38 activation (Lei et al. 2008)? This should be explained and for the impact of these signaling pathways on cell proliferation it would be nice if additional controls (e.g. kinase inhibitors) would be included.
  • The authors claim that genomic (TR) and non-genomic (T4-integrin-(ERK/p38?)) signaling contributes to the differentiation of the myoblasts. What about the other non-genomic pathway that is mentioned? Data on this signaling pathway need to be included to substantiate the authors’ conclusion.

Minor comments:

  • n-numbers need to be given for all experimental conditions and the type of statistic that was applied.
  • 1D: The time point of measurements needs to be given.
  • Statistics is not given in all figures. They need to be included. If no significant differences are proposed, “n.s.” should be indicated; otherwise, it is hard to follow the conclusions in the text, esp. since the results are often not that clear.
  • It is not clear if TH were added, e.g. in the αVβ3-Integrin experiments.
  • Why is the percentage of cell proliferation in Fig. 2 so much higher than in figure 1?

Author Response

We thank the reviewer for his/her positive comments. Enclosed is a point-by-point replay to each concern.

Reviewer 2 Report

Dear Editor,

The work presented in this manuscript is very interesting. The reviewer has some background in nuclear receptor biology. As the current understanding of how nuclear receptors work the widely accepted understanding is that nuclear receptor work mainly as transcription factors. On the other hand, several lines of evidence have been presented that nuclear receptors might act through non-cannonical mechanisms e.g by involvement in membrane-localized signal transduction pathways. There is a significant amount of data for estrogen receptors and some other nuclear receptors that enforce these alternative ways on how nuclear receptors might influence specific physiological processes.

The presented article is providing a high amount of data to underly how TR might influence the cellular mechanisms of muscle differentiation.

Any paper that is challenging current knowledge is difficult to publish. The authors present a high amount of scientific data to underly their hypothesis.

Accepting for publication of such a manuscript is always somehow risky as there might be other mechanisms to explain the findings. On the other hand, the presented experiments are well performed, presented clearly and only later follow-up experiments will be able to prove if the suggested mechanisms are the major ones that are explaining the finding.

Both the results and the presented explanations might result in later scientific discussions but the reviewer considers these discussions important for the progress of science.

Therefore I suggest the publication of the manuscript in the present form.

Author Response

REVIEWER 2

Dear Editor,

The work presented in this manuscript is very interesting. The reviewer has some background in nuclear receptor biology. As the current understanding of how nuclear receptors work the widely accepted understanding is that nuclear receptor work mainly as transcription factors. On the other hand, several lines of evidence have been presented that nuclear receptors might act through non-canonical mechanisms e.g. by involvement in membrane-localized signal transduction pathways. There is a significant amount of data for estrogen receptors and some other nuclear receptors that enforce these alternative ways on how nuclear receptors might influence specific physiological processes.

The presented article is providing a high amount of data to underly how TR might influence the cellular mechanisms of muscle differentiation.

Any paper that is challenging current knowledge is difficult to publish. The authors present a high amount of scientific data to underly their hypothesis.

Accepting for publication of such a manuscript is always somehow risky as there might be other mechanisms to explain the findings. On the other hand, the presented experiments are well performed, presented clearly and only later follow-up experiments will be able to prove if the suggested mechanisms are the major ones that are explaining the finding.

Both the results and the presented explanations might result in later scientific discussions, but the reviewer considers these discussions important for the progress of science.

Therefore, I suggest the publication of the manuscript in the present form.

We greatly appreciate the reviewer’s considerations and indeed, we agree that many aspects of the non-genomic actions of thyroid hormone action still remain to be explored. However, the study of TH effects in the muscle physiopathology will benefit from the current and the future approaches to fully address the complex mechanisms exerted by TH in such a relevant metabolic tissue.

Reviewer 3 Report

This work by Nappi et al. examine the role of Thyroid Hormone in muscle cells. The authors show ablation of D2 in C2C12 cells promotes proliferation . On the other hand, β3-KO cells show decreased cell proliferation. In addition, both D2 KO and β3-integrin KO block differentiation and antioxidant gene expression.

Overall this study is elegantly performed and well written,  the data convincing. Therefore, this reviewer recommends publication with some minor revisions outlined bellow.

  1. Figure 1E, statistical analysis is needed.
  2. Figure 1F, 2E, 5A-C, the text are too small to read.

Author Response

REVIEWER 3

This work by Nappi et al. examine the role of Thyroid Hormone in muscle cells. The authors show ablation of D2 in C2C12 cells promotes proliferation. On the other hand, β3-KO cells show decreased cell proliferation. In addition, both D2 KO and β3-Integrin KO block differentiation and antioxidant gene expression.

Overall, this study is elegantly performed and well written, the data convincing. Therefore, this reviewer recommends publication with some minor revisions outlined below.

Figure 1E, statistical analysis is needed.We apologize for not showing the statistic analysis properly.

We have included those analysis in the revised version of the manuscript.

Figure 1F, 2E, 5A-C, the text are too small to read.

The Figure size has been improved.

Round 2

Reviewer 1 Report

With regards to the "NLS" question:

Are there other examples of mutations that are able to override the nuclear translocation of an NLS-sequence? Did you remove the NLS or is it a “naturally” included NLS in the TRa and TRb?? Then, why do you call it NLS? You really need to rename the constructs so that it is clear that they are no real NLS-constructs anymore. It would be kind of you for the readers to clarify the changes in the genes (not just by naming the amino acids but also their function) and the naming of the constructs!

Author Response

Reviewer 1

Q. Are there other examples of mutations that are able to override the nuclear translocation of an NLS-sequence? Did you remove the NLS or is it a “naturally” included NLS in the TRa and TRb?? Then, why do you call it NLS? You really need to rename the constructs so that it is clear that they are no real NLS-constructs anymore. It would be kind of you for the readers to clarify the changes in the genes (not just by naming the amino acids but also their function) and the naming of the constructs!

A. The plasmids that we generated and called “TRa-NLS mutant and TRb-NLS mutant “ have point mutations in the nuclear translocation signal (NLS) of the wild type TRa and TRb genes. The mutation has been already described by Zhu XG et al, JBC 1998 (as cited in our manuscript, ref 36). This mutation causes the accumulation of the receptor in the cytoplasm and avoids the nuclear translocation. In order to simplify the general understanding of the constructs, we renamed them as D-mut (since the mutated sequence is in the D domain of the protein).